# Characteristics and Experiences of Ride-Hailing Drivers with Electric Vehicles

**Angela Sanguinetti** *[ID] **and Kenneth Kurani** [ID]

Plug-in Hybrid & Electric Vehicle Research Center, University of California, Davis, CA 95616, USA; knkurani@ucdavis.edu
* Correspondence: asanguinetti@ucdavis.edu

**Abstract:** Electrification of transportation network companies (TNCs), such as Uber and Lyft, can produce social and environmental benefits from reduced vehicle emissions and enhanced implementation of renewable electricity as well as private benefits to drivers via reduced vehicle fuel and maintenance costs compared to conventional vehicles. We conducted a survey of plug-in electric vehicle (PEV) drivers on the Uber platform in the US. This paper describes these drivers and their experiences to further understanding of motivations for and barriers to PEV adoption among TNC drivers. The TNC-PEV drivers in this sample clearly recognized, and were largely motivated by, economic benefits of fuel and maintenance savings, thus, increased net earnings, associated with using a PEV to provide ride-hailing services rather than a conventional internal combustion engine vehicle. Most drivers reported charging their PEV every day, most often at home and overnight. This is true even of those with plug-in hybrid electric vehicles (PHEVs) that can run on gas if not charged. Increased electric driving range topped the list of drivers' wishes to better support PEVs on TNCs, and range limitations topped the list of reasons why PHEV drivers did not opt for a battery electric vehicle (BEV; that runs exclusively on electricity). The second most common wish among all PEV drivers was for more charger locations.

**Keywords:** electric vehicle; consumer adoption; ride-hailing; transportation network company

## 1. Introduction

There is some evidence that the introduction of transportation network companies (TNCs), such as Uber and Lyft, has had positive environmental impacts through reduced vehicle ownership and emissions [1]. However, other research suggests TNCs may increase energy consumption [2], congestion [3], and vehicle-miles-traveled (VMT) [4]. George and Zafar [5] observed that these outcomes are likely context-dependent. For example, Alemi et al. [6] found that some TNC user segments increased the amount of their travel done in light-duty vehicles by shifting away from public transit or active modes to ride-hailing, while other TNC user segments decreased personal vehicle use by shifting toward public transit.

Regardless of the present net environmental impact of TNCs, industry stakeholders, policymakers, and academicians are predicting a future where shared mobility contributes to a low carbon transportation system [7–9]. In particular, a vision is converging around shared, automated, and electric right-sized vehicles integrated via multimodal MaaS platforms. Interim steps along the path to achieving this future include "electrifying TNCs" within the current model of conventional (non-automated) vehicles.

Emerging regulations are starting to shape this future. For example, California's recently enacted Senate Bill 1014 establishes a clean-miles standard for TNCs which requires that a growing percentage of ride-hailing services be provided by zero-emissions vehicles. Based on data from TNCs and charging infrastructure, Jenn [10] estimated that electrifying TNCs has potential for large greenhouse gas (GHG) emissions reductions.

Among car-buyers generally, awareness and knowledge, as well as consideration for purchase, of plug-in electric vehicles (PEVs) are low and unchanging over time despite increasingly positive economics for consumers, increasing makes and models available, and continued PEV charging deployment [11]. Barriers to adoption have included higher purchase price, limited vehicle range, and charging requirements (access and speed). Though the gap is decreasing, the purchase and lease costs of PEVs were still typically higher than comparable-size gasoline cars [12] at the time of this study. For some consumers and vehicle models, PEV purchase price is higher even after factoring in federal, state, and local incentives. Lower fuel and maintenance costs for PEVs may balance the higher upfront costs over time and even result in long-term savings in some cases. The fuel and maintenance cost advantages are higher with battery electric (i.e., all-electric) vehicles (BEVs) compared to plug-in hybrid electric vehicles (PHEVs) that can run on either gas or electricity.

Some of these PEV adoption barriers as well as benefits may be heightened for the TNC use case. For example, range limitations and time to charge (combined with limited and costly fast charging infrastructure) could be more problematic for TNC drivers who use their cars more intensively than the general population [5,13]. On the other hand, TNC-PEV drivers may reap greater benefits from fuel and maintenance savings and thus be more likely to achieve net cost savings [5,13]. However, factoring in costs for fast charging needs of BEVs used intensively for ride-hailing services, Pavlenko et al. [13] estimate that BEVs do not yet surpass hybrid vehicles in terms of economic advantage for ride-hailing use. They estimate that BEVs will reach cost parity with hybrids in 2023 (due to decrease in battery costs) for users who can charge at home at night, but not until after 2025 for users relying only on public charging, and thus conclude that in the interim, TNCs and policymakers will need to provide more affordable fast charging infrastructure to support TNC electrification.

George and Zafar [5] noted that current assumptions about the barriers to PEV adoption among ride-hailing drivers are largely anecdotal and called for more engagement with drivers to (a) understand the most significant barriers, (b) identify charging infrastructure needs, and (c) identify characteristics of those likely to adopt PEVs without additional incentives. The present research begins to address these needs. This is the first report from the 2019 North American Uber PEV Driver Survey. It describes the characteristics and experiences of PEV drivers on the Uber platform, including their demographics, motivations for adopting a PEV, charging behavior, and perceptions of priorities for expanding TNC electrification. A greater understanding of these should have policy implications in terms of reasonable assumptions and goals for TNC electric passenger vehicle-miles-traveled (eVMT), as well as implications for strategies to promote PEV adoption among ride-hailing drivers.

## 2. Materials and Methods

Researchers partnered with Uber to survey their drivers who use PEVs to provide ride-hailing services in North America (including the US and Canada). The survey was conducted in March and April 2019. The survey was distributed by Uber via email. All PEV drivers on the platform were recruited, but sampling ceased when a quota ($n = 400$) was achieved for US PHEV drivers since this is a much larger sub-population relative to US BEV drivers, Canadian PHEV drivers, and Canadian BEV drivers. (By the time sampling ceased, the quota was slightly exceeded; the number of US PHEV drivers in the sample is 415).

The final sample included 780 drivers, with 732 from the US and 48 from Canada. TNC drivers in California made up 47% of the US sample; 42 other US states/territories including Washington, D.C. and Puerto Rico were represented. Estimated response rate (population size undisclosed by the TNC) was 10%. Responses from Canadian drivers were excluded from these analyses due to the small sample of drivers who are subject to a different national-level policy and regulatory framework.

Survey questions (listed in Table 1) addressed five main topics:

1.  Who: Who (in terms of basic demographics) is currently driving PEVs on TNCs;

2. What: What PEVs they chose and how they acquired them;
3. Why: Motivations to drive a PEV for ride-hailing;
4. How: Ride-hail driving and PEV charging behaviors;
5. What drivers say would better support the use of PEVs for ride-hailing.

**Table 1.** TNC-PEV Driver Survey Questions.

| Category | Question Prompt | Response Options |
|---|---|---|
| Who | How old are you? | 21 to 29; 30 to 39; 40 to 49; 50 to 59; 60 to 69; 70 or older |
| | What is your gender? | Female; Male; Non-binary; Prefer not to answer; Self-describe: (open) |
| | What is the highest level of formal education you personally have completed? | Less than high school; High school or GRE; Some college/no degree; Associate's degree; Bachelor's degree; Graduate degree |
| | What was your household's pre-tax income from all sources for the past tax year? | Slider to select any number between $0 and $200,000 or more; No answer |
| | How would you describe the building in which you live? | Detached single family house; Duplex, row house or townhouse; Apartment or condominium; Mobile home, trailer, recreational vehicle, car, boat or other movable dwelling; Dorm room or fraternity/sorority house; Other |
| | Do you own, rent, or lease the building in which you live? | Own; Rent or lease; Employer-provided; Other |
| | How many people live in your household? | 1; 2; 3; 4; 5; 6; 7; 8 or more |
| What | What is the PEV that you drive on Uber? | Year, Make, Model, Option |
| | Do you own, lease, or rent this [PEV]? | Own, Lease, Rent, Other |
| | Is the [PEV] your first electric vehicle? | Yes, No |
| | Which came first for you, getting an electric vehicle or rideshare driving? | Electric vehicle before I drove rideshare; Rideshare driver before I got an electric vehicle; I am not sure which was first |
| Why | Why did you choose to drive an electric vehicle for rideshare instead of a conventional gas vehicle? | Choose up to three: To save money on fuel; To save money on maintenance; Longer lifetime of the vehicle; Better for the environment; Carpool lane access; Free or priority parking; Ability to drive for premium service (e.g., Uber Select); Other: |
| | Why did you choose to drive a battery electric vehicle for rideshare instead of a plug-in hybrid electric vehicle? (Chose up to three.) | Battery electric vehicle driving range long enough; I'm able to charge at home; Liked the vehicle body style; Enough charging infrastructure where I drive; Time away from driving to charge is not a problem; Not concerned about limiting the rides I could take; To increase my profits from driving on Uber; Other: |
| | Why did you choose to drive a plug-in hybrid electric vehicle for rideshare instead of a battery/all electric vehicle? (Choose up to three.) | Did not want to be nervous about range; Unable to charge vehicle at home; Vehicle body style I wanted not available in a battery electric vehicle; Lack of charging infrastructure where I drive; Didn't want to take time away from driving to charge; Didn't want to limit rides I could take; To increase my profits from driving on Uber; None of these; Other: |
| | When you bought/leased/rented/acquired your [PEV] did you compare all costs (purchase, fuel, maintenance, insurance, etc.) of it compared to other vehicles? | Yes; No; Don't recall |
| | Do you compare all such costs for any car you buy or lease? | Always; Sometimes; Never |
| | Would you still drive for rideshare if you could only have a conventional gas vehicle? | Yes; No; Not sure |
| | Would you still drive an electric vehicle if you did not drive on rideshare networks? | Yes; No; Not sure |

**Table 1.** *Cont.*

| Category | Question Prompt | Response Options |
|---|---|---|
| | Do you drive for more than one rideshare network (Uber, Lyft, or others)? | Yes; No |
| | Besides your [PEV], is there any other vehicle you are presently driving on rideshare networks? | No, only my [PEV]; Yes, I sometimes drive a different vehicle for rideshare |
| | About how many hours per week do you drive on rideshare networks (Uber, Lyft, and similar)? | Ten or fewer hours; 11–20 h; 21–30 h; 31–40 h; 41 or more hours |
| | On how many days do you charge your [PEV]? | I don't charge my [PEV]; Less than once a month; More than once a month, but less than weekly; One or two days a week; Three or four days a week; Five or six days a week; Every day |
| | What percent of all your charging of your [PEV] do you do at home, at public chargers, or at a workplace? | Percentage for each: home, public, workplace; totaling 100 |
| | You've indicated you don't charge your [PEV] at home. Could you charge it at home if you wanted to? | Yes; No; I don't know |
| | What percent of all your charging of your [PEV] do you do at different power levels or rates? | Percentage for each: fast, level 1, and level 2 chargers; totaling 100 |
| How | How much of all your charging of your [PEV] do you do during different periods of the day? | Percentage for each: midnight to 6 a.m. 6 a.m. to noon, noon to 6 p.m., and 6 p.m. to midnight; totaling 100 |
| | What is the largest number of times you will charge your [PEV] during any single period of driving for ridesharing? | None; Once; Twice; Three times or more |
| | Please estimate the time it takes for each of these steps in charging your [PEV]: Time to reach charger location from the moment you decide to charge; Time waiting in line to charge; Time to actually charge your vehicle; Time to get your next ride after charging | Shortest time is about XXX minutes; Longest time is about XXX minutes |
| | Do you feel you miss out on earning opportunities because you have to charge your [PEV]? If so, how often do you feel the missed opportunity? | Not at all; Once per 15 charges or less; Once per 10 charges; Once per 5 charges or more frequent |
| | Do you do any of the following to increase your electric driving range while driving for ridesharing? | Limit top speeds; Moderate accelerations; Try to coast more to stops; Choose trips or routes where I can use less energy; Choose trips or routes where I can use more regenerative braking; Use heating or air conditioning less; Leave more room from car ahead; None of the above; Other: |
| | How would you improve your experience as a rideshare driver with an electric vehicle? | Choose up to three answers: Longer electric driving range; More charger locations; Reliable real-time information about charger availability; More reliable information on vehicle driving range; Additional trip destination setting tokens to and from charging stations; More detailed trip information in advance of trip acceptance; Financial bonuses for completing complete trip target numbers; More passenger seats; More cargo space; Other: |

## 3. Results

### 3.1. Who Is Currently Driving PEVs on TNCs

Table 2 describes this sample of TNC-PEV drivers. A large majority were male and median age was 40–49 years (Figure 1). A large majority (90%) had at least some college-level education. Median reported household income was $88,000 (treating "$200,000 or more" as $200,000; Figure 2). Mean household size was three members (treating "8 or more" household members as 8). Most were living in a single-family home and a slight majority were homeowners.

**Table 2.** TNC-PEV Driver Characteristics.

| Variable | Descriptive Statistics |
| --- | --- |
| Age | *Mdn* = 40 to 49 years old; 9% 21 to 29, 27% 30 to 39, 28% 40 to 49, 21% 50 to 59, 10% 60 to 69, 4% 70 or older |
| Gender | 85% Male, 13% Female, 1% Non-binary, 1% Prefer not to answer |
| Education | *Mdn* = Bachelor's degree; <1% Less than high school, 9% High school or GRE, 27% Some college/no degree, 11% Associate's degree, 34% Bachelor's degree, 18% Graduate degree |
| Income | *Mdn* = $88,000; min = $1000, max = $200,000 or more; 22% no answer |
| Household Size | *Mdn* = 3; 15% 1, 31% 2, 20% 3, 18% 4, 12% 5, 3% 6, 1% 7, 1% 8 or more |
| Housing Type | 62% Detached single-family home, 24% Apartment/condo, 12% Duplex/row house/townhouse, 2.5% Mobile/dorm/other |
| Housing Tenure | 55% Own, 43% Rent, 2% Other |

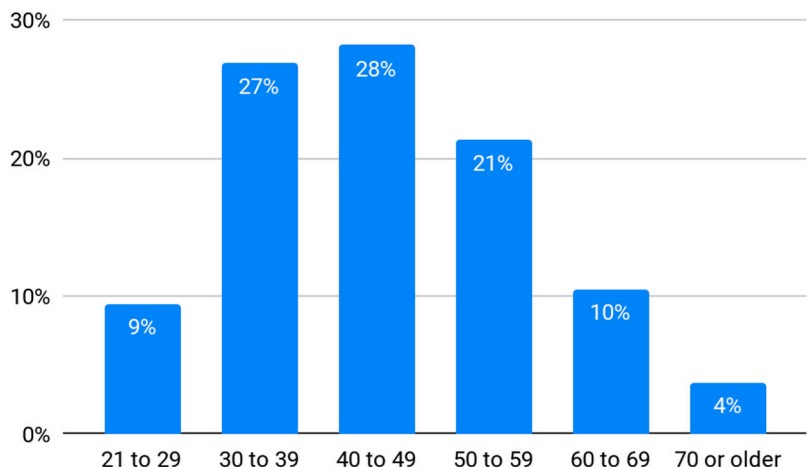

**Figure 1.** Age distribution of TNC-PEV drivers.

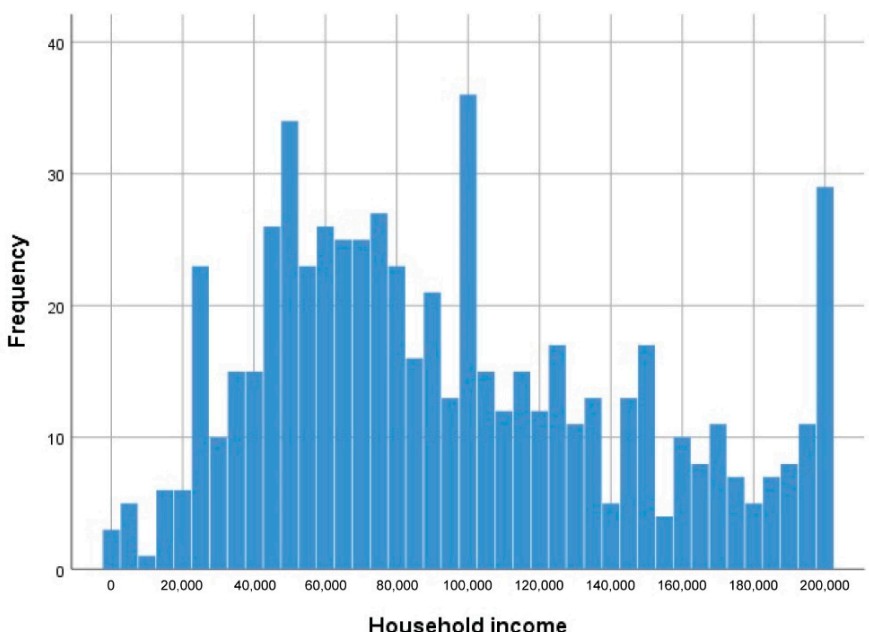

**Figure 2.** Income distribution of TNC-PEV drivers.

### 3.2. What Cars They Chose and How They Acquired Them

Figure 3 presents the most common PEVs in the sample. The three most common BEVs were the Chevrolet Bolt, Tesla Model S, and Nissan Leaf. The three most common PHEVs were the Ford Fusion Energi, Toyota Prius Plug-in, and Volt (C-MAX Energi close 4th). (The Tesla Model 3 and Toyota Prius Prime were too recent to be present in large numbers in the sample.) While 30 makes and models of PEVs were in the sample, the top five accounted for two-thirds of the sample; the top ten accounted for 93%.

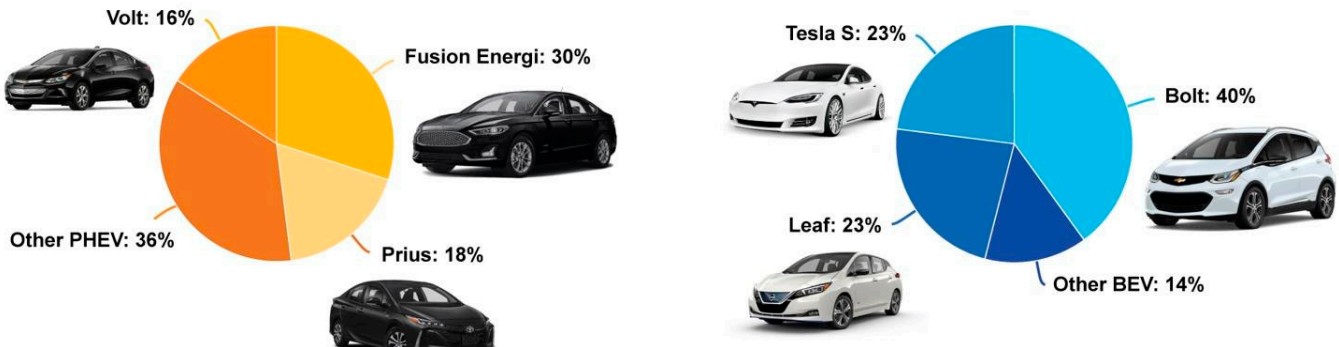

**Figure 3.** Most common PHEV and BEV models used by TNC drivers.

The distributions of electric range in our sample were:

- BEV: Range = 50 to 300 miles; mean = 190 miles; median = 210 miles;
- PHEV: Range = 3 to 100 miles; mean = 30 miles; median = 25 miles.

The high median range for BEVs emphasizes the extent to which Chevrolet Bolt EV and Tesla Model S shape the BEV range distribution.

For 82% of respondents, the PEV they were driving on TNCs was their first PEV. There is an approximately even split between those who started ride-hail driving before (47% ride-hail driver first) versus after (50% PEV driver first) acquiring a PEV (3% not sure). Most drivers (73%) owned their PEV (16% leased and 11% rented). We did not ask drivers directly whether they participate in the Maven weekly car rental program; vehicles for rent include Chevrolet Bolt EVs. We presume that PEV drivers who drove a rented Chevrolet Bolt EV in a city where Maven was available were driving a PEV from Maven.

### 3.3. Why Drive a PEV for Ride-Hailing?

Figures 4–6 present reasons drivers selected from a list regarding their decision to drive a PEV rather than an internal combustion engine vehicle (ICEV) for ride-hailing, and reasons for choosing either a BEV or PHEV over the other. (The lists of potential reasons were generated during prior focus groups with people driving PEVs on TNCs).

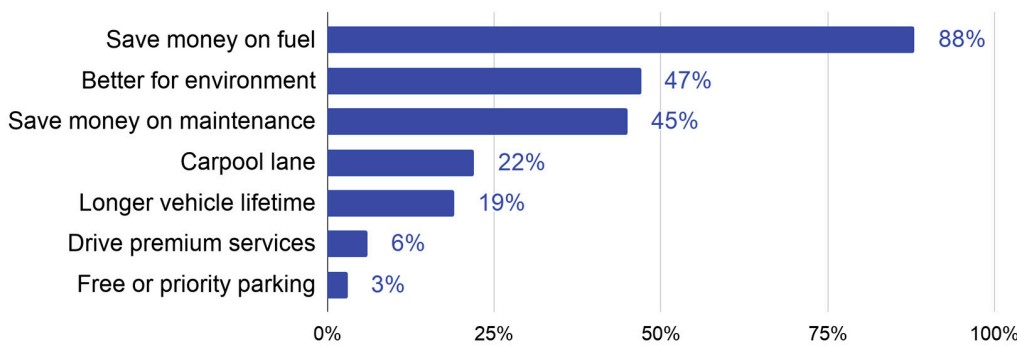

**Figure 4.** Reasons for choosing a PEV over an ICEV. Multiple answers allowed.

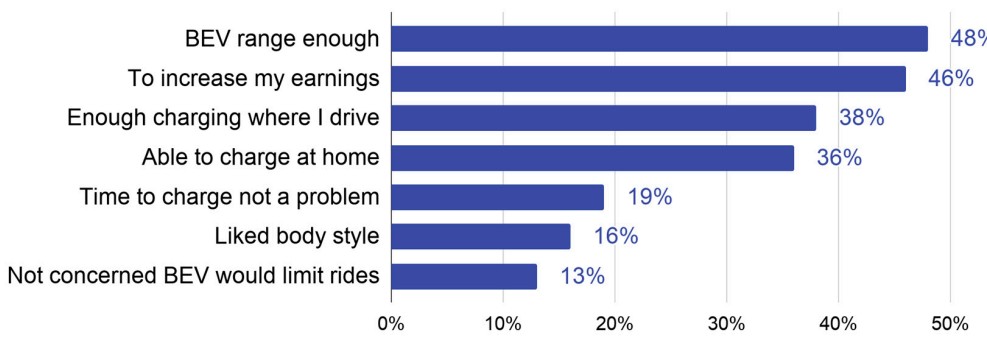

**Figure 5.** Reasons for choosing a BEV over a PHEV. Multiple answers allowed.

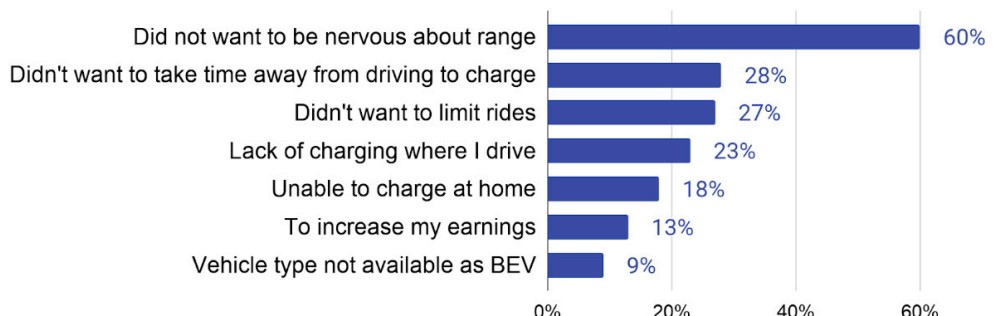

**Figure 6.** Reasons for choosing a PHEV over a BEV. Multiple answers allowed.

Saving money on fueling costs was by far the most oft-cited reason for choosing a PEV rather than an ICEV. Saving money on maintenance costs was third, following very closely after driving a car that is better for the environment. Further supporting the importance of financial motivations, 72% said they compared all costs for their PEV to other vehicles; of these, 74.5% reported they always do this with any vehicle they acquire.

Common reasons for choosing a BEV over a PHEV to drive on TNCs were for greater earnings and because BEV driving range and access to charging were adequate for their needs (Figure 5). In direct contrast, "range anxiety" was by far the most commonly selected reason for choosing a PHEV over a BEV to drive on TNCs (Figure 6). In addition, several participants wrote open-ended comments indicating they were certain that BEV range would be insufficient because they regularly drove long distances. Nineteen participants mentioned cost barriers in open-ended comments, mainly that they chose a PHEV because upfront costs were too high for desirable BEVs. Notably, these cost concerns were unaccounted for in closed-ended response options, so the prevalence of cost barriers should be more accurately estimated in future studies.

Most (90%) reported that they would still drive a PEV even if they didn't drive for ride-hailing services (6% No, 4% Not sure); see Figure 7. Only 41% reported the opposite: that they would drive on TNCs if they had an ICEV (35% No, 24% Not sure).

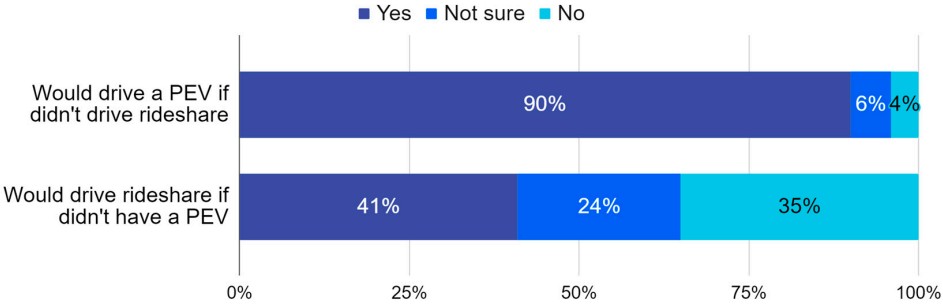

**Figure 7.** Commitment to PEV relative to commitment to ride-hailing job.

### 3.4. Ride-Hail Driving and Charging Behaviors

Most drivers reported that they drive for more than one TNC (65%) and that their PEV is the only car they use when driving on TNCs (76%); 24% reported that they also drive another vehicle for ride-hailing. Reported hours spent driving on TNCs per week is relatively evenly distributed across four categories ranging from "up to 10" to "40 or more" hours (Figure 8). A majority (52% of PHEV drivers and 65% of BEV drivers) reported they charge their PEV every day (Figure 9). However, 11% of PHEV drivers report they never charge their car.

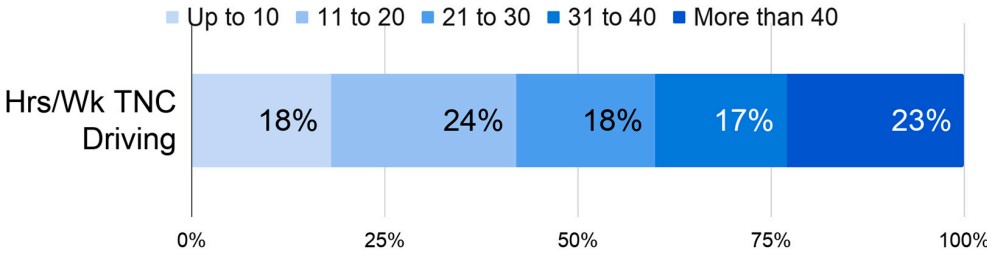

**Figure 8.** Average number of ride-hail driving hours per week for aggregate sample of drivers.

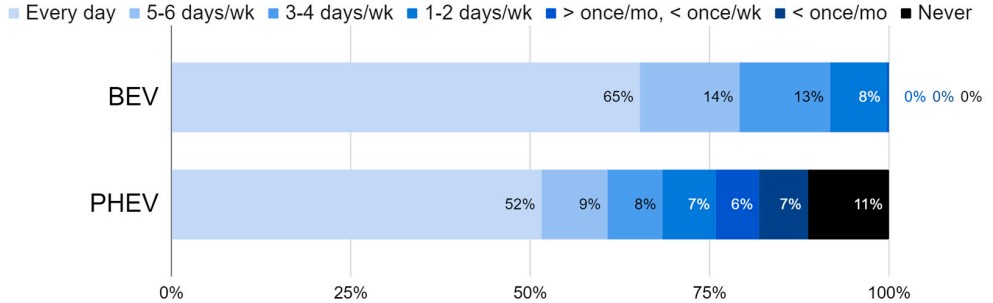

**Figure 9.** Charging frequency for BEV and PHEV TNC drivers.

On average, TNC-PEV drivers do most of their charging at home (58% of the time on average; Figure 10). However, nearly one-quarter (23%) said they never charge at home; of these, more than half (54%) said they were unable to charge at home, 39% said they could, and 7% did not know. Drivers reported charging most often between 12 a.m. and 6 a.m. (Figure 11). Combined with charging from 6 p.m. to midnight, 70% of charging is done during the evening and night.

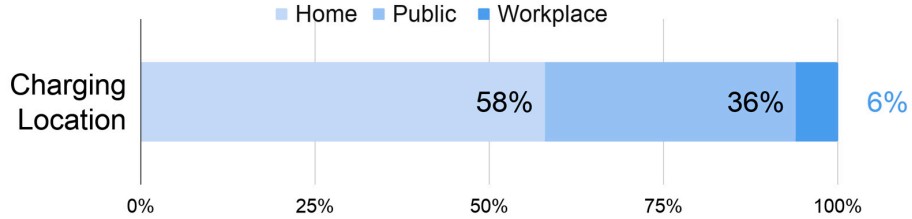

**Figure 10.** Mean distribution of charging location.

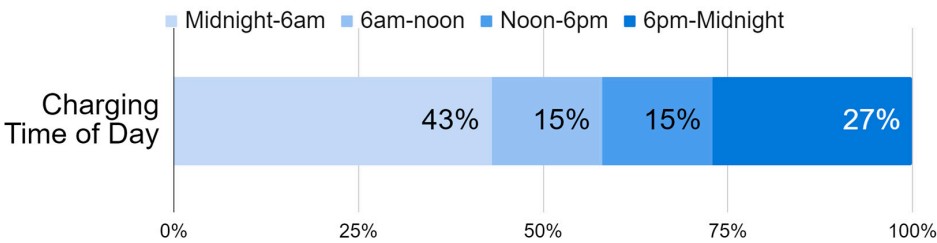

**Figure 11.** Mean distribution of charging time of day.

We conducted k-means cluster analysis to assess whether TNC-PEV drivers could be categorized into groups by their use of different power levels to charge their PEVs. The seven-cluster solution is depicted in Figure 12. The mean percentage of each level of charging power is illustrated for each cluster. One of these is a cluster for TNC-PEV drivers who do not charge ("None"); thus, its mean at all three power levels is 0%.

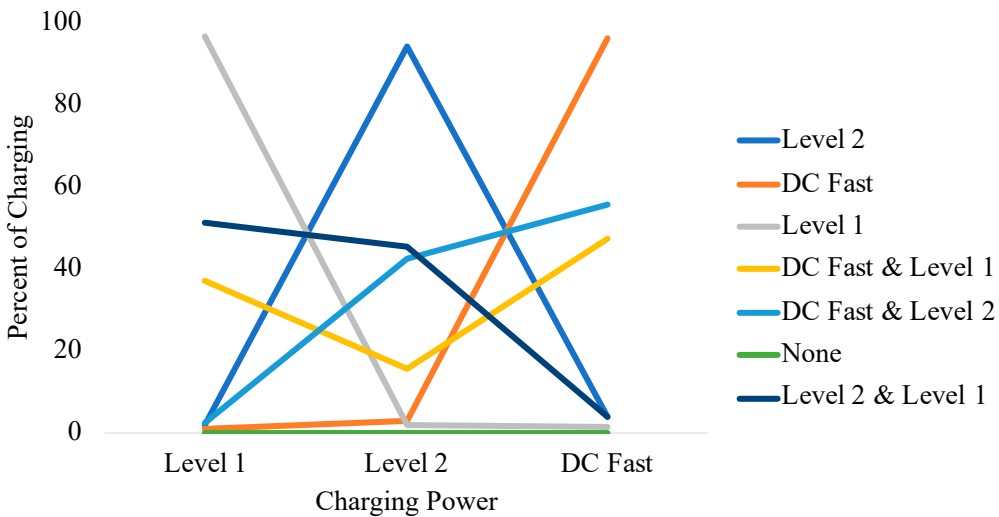

**Figure 12.** Distribution of mean charging levels for seven driver clusters.

The majority of drivers (78%) belong to the first three "single-power" level clusters; that is, most TNC-PEV drivers do most of their charging at a single power level (Figure 13). Forty-four percent of drivers (*n* = 319) owned a PEV that can DC fast charge. Of these, 90% reportedly do nearly all their charging ("DC fast") or about half their charging ("DC Fast & Level 1" and "DC Fast & and Level 2") at DC fast chargers. Conversely, few TNC-PEV drivers who own a vehicle capable of DC fast charging do not use fast charging at all.

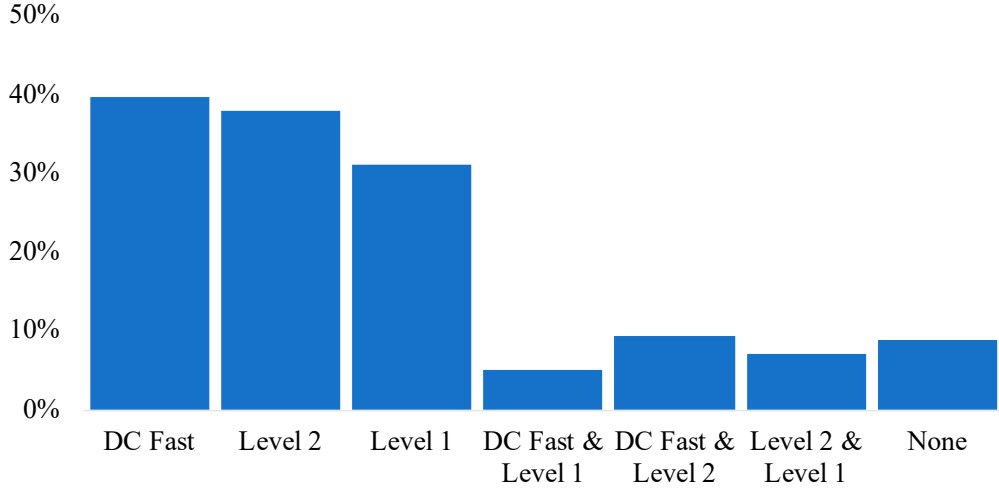

**Figure 13.** Distribution of TNC-PEV drivers across seven charging power clusters.

We also asked drivers for the maximum number of times they will charge per "shift", i.e., episode of ride-hail driving. "Once" was the most common response (41%), followed by None (24%), Twice (23%), and Three times or more (11.5%). Number of charges per "shift" was correlated with hours spent ride-hail driving per week (*r* = 0.23, *p* < 0.001). For those who reported charging during TNC "shifts", we asked how long various aspects of the charging process take; here are the averages:

- Time to reach charger: 16–47 min;

- Time waiting in line to charge: 6–28 min;
- Time to charge: 73–176 min;
- Time to next fare: 8–42 min.

Of these drivers, most (54%) reported that they do not miss out on earning fares due to charging; others reported that they do miss out at least once every 5 charges (24%), once per 10 charges (10%) or once per 15 charges or less (12%).

Most drivers reported engaging in at least one strategy to maximize their electric range, e.g., moderate acceleration (67%), coast more (55%), limit top speeds (52%), use less HVAC (39%), or increase following distance (25%).

### 3.5. Driver Ideas to Support the Use of PEVs for Ride-Hailing

Figure 14 presents drivers' rankings of the top three strategies they thought would improve their experience driving PEVs on TNCs, from a list of provided options. Most drivers ranked longer electric range and more charger locations in their top three potential improvements. Financial bonuses for meeting trip targets and more pre-trip information were also popular choices. Despite the common use of charging while driving on TNCs among these drivers, only one-in-five mentioned reliable real-time charger information.

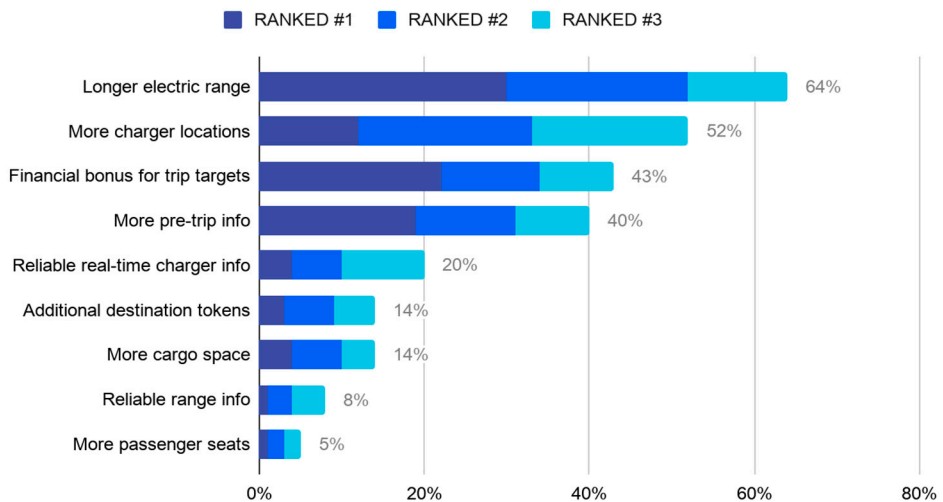

**Figure 14.** Driver choices to improve the experience of driving PEVs on TNCs; up to three answers allowed.

### 4. Discussion

This report describes a large sample of TNC-PEV drivers in the US in terms of their demographic characteristics, motivations for driving PEVs on TNCs, charging while driving on TNCs, and their ideas to improve the experience of driving PEVs on TNCs. TNC-PEV drivers are:

- Much more likely to be male than female;
- More likely to live in a household of two or three people in a single-family home they own rather than any other building type or residential tenure;
- Very likely to be between 30 and 60 years old;
- More likely than not to have some level of collegiate education culminating in a Bachelor's or graduate degree;
- More likely than not to have a household income less than $100,000.

Further, in terms of their PEV, these TNC drivers are:

- More likely to drive a PHEV than a BEV;
- Much more likely to own their PEV than to lease or rent;
- Most likely to be driving their first PEV;
- Not likely to drive any other vehicle to provide ride-hailing services.

The TNC-PEV drivers in this sample clearly recognized, and were largely motivated by, economic benefits of fuel and maintenance savings, thus, increased net earnings, associated with using a PEV to provide ride-hailing services rather than an ICEV. Further research is required to understand TNC drivers' perceptions of total costs of ownership for different vehicle drivetrains and differences in these perceptions between drivers of BEVs, PHEVs, hybrids, and ICEVs. Environmental motivations for PEV adoption were also quite common. TNC-PEV drivers are typically more committed to continuing to drive a PEV than to continuing to drive on TNCs.

Most drivers are charging their PEV every day, most often at home and overnight. This is true even of those with PHEVs. While most say they are willing to charge once or more while actively driving on TNCs, one-fourth say they are not willing to do so. As DC fast charging is by definition "public charging", i.e., no one can DC fast charge at home, the presence of a large cluster of TNC-PEV drivers who do, on average, half or nearly all of their charging at DC fast chargers suggests these PEV drivers are especially reliant on publicly-available, DC fast charging. Subsequent analysis of these data will further explore charging behavior based on driver and vehicle characteristics. For example, Wenzel et al. [2] estimated that 90% of ride-hailing shifts could be accomplished on a single charge with 200-mile range BEVs, so drivers with long-range BEVs likely have a very different charging profile compared to shorter range BEV and PHEV drivers.

Increased range topped the list of PEV drivers' wishes to better support PEVs on TNCs, and range limitations topped the list of reasons why PHEV drivers did not opt for a BEV. The next most common wish among all PEV drivers was for more charger locations. The third and fourth ranked wishes were financial bonuses for trip targets and more pre-trip information, factors more controllable by TNCs. These findings suggest priorities for policies to support TNC electrification, both for regulators and for TNCs directly.

This first report from the 2019 North American Uber PEV Driver Survey summarized the characteristics and experiences of these drivers, in aggregate. Further analyses will explore relationships between driver characteristics, behaviors (e.g., amount of time providing ride-hailing services and charging practices), PEV type, electric range, and vehicle acquisition models. Subsequent analyses will also draw on other datasets to compare this group of TNC-PEV drivers to the general population of PEV drivers and to TNC drivers who do not use PEVs. These comparative analyses will yield further implications for the prospects of TNC electrification.

**Author Contributions:** Conceptualization, A.S. and K.K.; methodology, A.S. and K.K.; formal analysis, A.S. and K.K.; writing-original draft preparation, A.S. and K.K.; review and editing, A.S.; project administration, A.S. and K.K. All authors have read and agreed to the published version of the manuscript.

**Funding:** This research received funding from Uber Technologies, Inc. (Uber).

**Acknowledgments:** The researchers are grateful to Uber for facilitating this work. Uber hosted focus groups with PEV drivers on their platform (an initial step to questionnaire design), sent the survey recruitment invitation to their drivers with PEVs, and paid for the translation of the questionnaire into French for their drivers in Quebec, Canada. Uber facilitated participant recruitment and provided feedback on various aspects of the study but had no right-of-refusal in study design, survey analyses, or interpretation of data. Uber had no role in writing of the manuscript or the decision to publish the results.

**Conflicts of Interest:** The authors declare they have no conflicts of interest.

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
