# Peer review of "Characteristics and Experiences of Ride-Hailing Drivers with Electric Vehicles"

_wevj, doi:10.3390/wevj12020079_

Round 1

Reviewer 1 Report

This is a good summary of a survey of TNC drivers who drive PEVs for a TNC service. The survey provides new information on driver demographics and driving shifts, their reasons for driving PEVs, and reasons PHEV drivers do not drive a BEV. The results would be more valuable if the survey included a control sample of non-PEV TNC drivers, to compare demographics, driving shifts, and reasons for PEV adoption of PEV with non-PEV drivers for TNCs.

I have only two very minor changes to the manuscript highlighted in the attached.

Author Response

The errors highlighted in the reviewer's file have been corrected. We agree a control sample would be valuable and are hoping to add that to a future study.

Reviewer 2 Report

The electrification of the transport network based on renewable electricity is a very important and up to date issue. Many companies and private owners are introducing Plug-in Hybrid Electric Vehicles (PHEV) or Battery Electric Vehicles (BEVs) into the transport network. The article describes the preferences for PHEV and BEV vehicles and summarizes the results of the survey questions. 

Author Response

We thank the reviewer for their time and support

Reviewer 3 Report

    This paper analyzes the characteristics and experiences of ride-hailing drivers with electric vehicles. The contents are very simple, and the following problems need to be revised.

--> Please describe the contributions of this study in detail in the section of “Introduction”.

--> Please review the existing researches regarding the electric ride-hailing service, and then summarize the research gaps.

--> Please supplement the contents of the questionnaire, such as the questions, the options for the questions.

--> Please list the basic attributes of the participants in the survey in a table.

--> Please compare the difference of the characteristics and experiences for ride-hailing drivers with BEV and PHEV, not just illustrate the reasons for choosing these two vehicles.

--> Please supplement the policy implications to the actual practice according to the results.

--> Please supplement some more references to support this study.

Author Response

We thank the reviewer for their attention and feedback. We recognize that the analyses are relatively simple but these simple data have value because they are the first of their kind. No other basic descriptive accounts of TNC PEV drivers are available and this is a large representative sample. 

--> Please describe the contributions of this study in detail in the section of “Introduction”...Please review the existing researches regarding the electric ride-hailing service, and then summarize the research gaps.

We believe we have done this. The main gap we are filling is the lack of data about TNC PEV drivers and particularly from TNC PEV drivers. If there are particular studies the reviewer notes missing, please share and we will be happy to include. We think we have captured all the most pertinent literature.

--> Please supplement the contents of the questionnaire, such as the questions, the options for the questions.

We have added a table with all the questions and response options for the data used in this manuscript. It is quite long, so the journal editors may wish to make it an Appendix, which would be fine with us as well.

--> Please list the basic attributes of the participants in the survey in a table.

We have added this table.

--> Please compare the difference of the characteristics and experiences for ride-hailing drivers with BEV and PHEV, not just illustrate the reasons for choosing these two vehicles.

This is beyond the intended scope of this manuscript. We realize that there are many important distinctions among the TNC PEV drivers in our sample, but a simple distinction between BEV and PHEV drivers within each of the analyses we provide in this paper would not be a sufficient solution. For example, we also need to consider BEV renters separately as they are likely quite unique from BEV owners/leasers (drive more hours on TNCs, may be younger, all drive a Bolt and acquired it after starting on TNCs). And within and across PHEVs and BEVs, the electric range of PEVs has many implications for drivers' charging practices. We have work forthcoming that covers these analyses, but first there is value in disseminating the aggregate results for all TNC PEV drivers. 

--> Please supplement the policy implications to the actual practice according to the results.

We have added a more explicit statement about policy implications

--> Please supplement some more references to support this study.

Again, we feel we included the pertinent sources, but please advise if there are some you note missing. We definitely aspire to be thorough.

Thank you again for your time.